# Continuously-Tempered PDMP samplers

**Matthew Sutton**
Centre for Data Science
Queensland University of Technology
`matt.sutton@qut.edu.au`

**Robert Salomone**
Centre for Data Science
Queensland University of Technology
`robert.salomone@qut.edu.au`

**Augustin Chevallier**
Department of Mathematics and Statistics
Lancaster University
`a.chevallier@lancaster.ac.uk`

**Paul Fearnhead** *
Department of Mathematics and Statistics
Lancaster University
`p.fearnhead@lancaster.ac.uk`

## Abstract

New sampling algorithms based on simulating continuous-time stochastic processes called piecewise deterministic Markov processes (PDMPs) have shown considerable promise. However, these methods can struggle to sample from multimodal or heavy-tailed distributions. We show how tempering ideas can improve the mixing of PDMPs in such cases. We introduce an extended distribution defined over the state of the posterior distribution and an inverse temperature, which interpolates between a tractable distribution when the inverse temperature is 0 and the posterior when the inverse temperature is 1. The marginal distribution of the inverse temperature is a mixture of a continuous distribution on $[0, 1)$ and a point mass at 1: which means that we obtain samples when the inverse temperature is 1, and these are draws from the posterior, but sampling algorithms will also explore distributions at lower temperatures which will improve mixing. We show how PDMPs, and particularly the Zig-Zag sampler, can be implemented to sample from such an extended distribution. The resulting algorithm is easy to implement and we show empirically that it can outperform existing PDMP-based samplers on challenging multimodal posteriors.

## 1 Introduction

Recently there has been considerable interest in developing sampling algorithms based on simulating continuous time stochastic processes called piecewise deterministic processes (PDMPs) Davis [1993]. These sampling algorithms are qualitatively similar to Hamiltonian Monte Carlo algorithms Neal [2011]. To simulate from some target distribution $\pi$, they work with an augmented state-space $(\boldsymbol{x}, \boldsymbol{v})$, where the $\boldsymbol{x}$ component can be viewed as position, and the $\boldsymbol{v}$ component as a velocity. The motivation for this is that these processes will encourage exploration of $\pi$ by simulating continuous velocity paths in between random event times at which the velocity changes. Examples of such sampling algorithms include the Bouncy Particle Sampler [Peters and de With, 2012, Bouchard-Côté et al., 2018], the Zig-Zag algorithm [Bierkens et al., 2019], the Coordinate Sampler [Wu and Robert, 2020] and the Boomerang Sampler [Bierkens et al., 2020]. See Fearnhead et al. [2018] for an overview of these methods. Importantly, it is simple to write down the event rate, and how the velocity should change at each event, to ensure these PDMPs have $\pi$ as their invariant distribution. These samplers depend on $\pi$ only through the gradient of $\log \pi$, and thus $\pi$ only needs to be known up to a constant of proportionality.

---

*The authors acknowledge funding through EPSRC grants EP/R018561/1 and EP/R034710/1

PDMP samplers have a number of advantages, including non-reversible dynamics (which are known to improve mixing relative to reversible processes [Diaconis et al., 2000, Bierkens, 2016]), and the ability to reduce computation-per-iteration by either leveraging sparsity structure in the model [Bouchard-Côté et al., 2018, Sutton and Fearnhead, 2021] or using only sub-samples of the data to approximate the log-likelihood at each iteration (whilst still guaranteeing sampling from the target [Bierkens et al., 2019]). However, like other MCMC algorithms, particularly those that use gradient information, these PDMP samplers can struggle to mix for multi-modal target distributions, or for heavy-tailed targets [Vasdekis and Roberts, 2021].

One of the more successful techniques for enabling an MCMC algorithm to sample from challenging, e.g. multi-modal, target distributions is to use tempering. There are various forms of tempering, but each is based on defining either a discrete set or continuum of distributions that interpolate between a distribution that is simple to sample from (viewed as at high temperature) and the target distribution (at a low temperature). The idea is that allowing moving across this set will improve mixing, as moving between modes will be easier for the distributions at higher temperatures. Examples of such algorithms include parallel tempering [Swendsen and Wang, 1986], simulated tempering [Marinari and Parisi, 1992], and continuous tempering [Graham and Storkey, 2017].

In this paper we show how continuous tempering ideas can be used with PDMP samplers. We have chosen *continuous* tempering, as opposed to the alternative tempering approaches, as it is the method that can most naturally benefit from the continuous-time nature of PDMP dynamics. To the best of our knowledge, this is the first attempt at using tempering ideas to improve PDMP samplers. The general idea is to define a joint distribution on the state of the PDMP, $z = (x, v)$, and the inverse temperature, $\beta$. The target distribution of interest is the $x$-marginal of this joint distribution when $\beta = 1$. We define the joint distribution so that it has a point mass at $\beta = 1$ — thus simulating from it will lead to a proportion of the resulting samples being from the target. We then use a PDMP sampler to simulate from this joint distribution. Constructing the appropriate dynamics of the PDMP sampler is non-trivial in this case as we have to deal with its behaviour as it transitions between the continuous distribution on $\beta \in [0, 1)$ and the point mass at $\beta = 1$. We use recent ideas for PDMP samplers with discontinuities [Chevallier et al., 2021, 2020] to solve this challenge.

While we express ideas based on and similar to Graham and Storkey [2017], a key difference is the inclusion of a point-mass at $\beta = 1$, this means we can obtain samples from $\pi$ rather than having to resort to importance sampling to correct samples drawn at different temperatures. It is easy to introduce a point mass into the dynamics of the PDMP sampler due to its continuous sample paths: we simulate paths for $\beta \in [0, 1)$ until the process hits $\beta = 1$ — we then simulate paths with $\beta$ fixed to 1 for an exponentially-distributed period of time before returning to $\beta \in [0, 1)$. By comparison, using a point-mass within a Hamiltonian Monte Carlo sampler is not possible as the discretised sample paths will not hit $\beta = 1$ with probability 1. Yao et al. [2020] consider an *indirect* approach to sampling with a point-mass at $\beta = 1$ using a continuous link function. Our approach is distinct and more direct, allowing one to exploit the unique advantages of PDMP samplers — such as the ability to perform subsampling without altering the ergodic distribution and application to sampling transdimensional distributions.

The benefits of introducing the point mass at $\beta = 1$ are numerically investigated, and practical considerations related to choosing tuning parameters to encourage a desired proportion of time at the target distribution are presented (Section 3.4). We find that, analogously to standard methods in discrete-time, the tempered counterparts of PDMP samplers outperform vanilla PDMP on challenging sampling problems.

## 2   The Zig-Zag sampler

For the purposes of brevity and ease of exposition, we focus specifically on a continuous-tempered version of the Zig-Zag sampler. However, we stress that the underlying ideas can easily be applied to *any* PDMP sampler, for example, the Bouncy Particle Sampler [Bouchard-Côté et al., 2018] or the Boomerang Sampler [Bierkens et al., 2020]; see the comments at the end of Section 3.2.

Consider the problem of sampling from a target density defined for $x \in \mathcal{X} := \mathbb{R}^d$ by

$$\pi(x) = \frac{1}{Z} \exp(-U(x))$$

where $U : \mathcal{X} \to \mathbb{R}$ is a differentiable function referred to as the potential and $Z$ is the, potentially unknown, normalising constant $Z = \int_{\mathcal{X}} \exp(-U(\boldsymbol{x}))d\boldsymbol{x}$. We will denote the un-normalised target density by $q$, so $\pi(\boldsymbol{x}) = q(\boldsymbol{x})/Z$.

The *Zig-Zag* process is a continuous-time PDMP, which can be defined so as to have $\pi$ as its invariant distribution. The process is defined on an extended state-space that can be viewed as consisting of a position $\boldsymbol{x}$ and a velocity component $\boldsymbol{v}$. For the Zig-Zag process, the velocity is restricted to be $\pm 1$ in each axis direction. Thus the extended space is $E = \mathbb{R}^d \times \{-1, 1\}^d$. We write $\boldsymbol{z} = (\boldsymbol{x}, \boldsymbol{v})$ for $\boldsymbol{z} \in E$ from here on. We use subscripts to denote time, and superscripts to denote components. So $\boldsymbol{z}_t$ will be the state at time $t$, while $\boldsymbol{x}_t^i$ will be the $i$th component of the position at time $t$.

For an event at time $t$ we use the notation $\boldsymbol{z}_{t-}$ for the state immediately before the event, and $\boldsymbol{z}_t$ the state immediately after it. The dynamics of the Zig-Zag process are deterministic between random event times. At each event time the direction of one component of the velocity is switched. The deterministic dynamics are specified by a constant velocity model. So, if there are no events between times $t$ and $t + h$, for $h > 0$, the change of state is given by $\boldsymbol{z}_{t+h} = (\boldsymbol{x}_{t+h}, \boldsymbol{v}_{t+h}) = (\boldsymbol{x}_t + h\boldsymbol{v}_t, \boldsymbol{v}_t)$.

The events occur with a rate that depends on the current state. For the Zig-Zag, process we have $d$ types of event, each of which results in the flipping of one of the $d$ components of the velocity process. To ensure that we have $\pi$ as the $\boldsymbol{x}$-marginal of the process's invariant distribution, these rates are defined to be, for $i = 1, \ldots, d$,

$$\lambda_i(\boldsymbol{x}_t, \boldsymbol{v}_t) = \max(0, \boldsymbol{v}_t^i \partial_{\boldsymbol{x}^i} U(\boldsymbol{x})),$$

with the transition at the corresponding event being that $\boldsymbol{v}_t^i = -\boldsymbol{v}_{t-}^i$, and all other elements of the state are unchanged.

Pseudo-code for simulating the Zig-Zag process is given in Algorithm 1. When we simulate such a process, the output of the algorithm is the set of event times, positions and velocities after the event times. This set is known as the PDMP skeleton and with the deterministic dynamics of the process it defines the continuous time path of the process. We can use such a simulated path to give us draws from $\pi(\boldsymbol{x})$, by discarding some suitably chosen initial path time as burn-in, and then evaluating the $\boldsymbol{x}$ component of the process at a set of evenly spaced discrete time points (see e.g. Fearnhead et al. [2018] for alternative approaches that use the continuous-time paths).

---

**Algorithm 1** Zig-Zag algorithm

1: **Inputs:** initial state $(\boldsymbol{x}_{t_0}, \boldsymbol{v}_{t_0})$ and number of simulated events $K$
2: $t_0 \leftarrow 0$
3: **for** $k \in 1, \ldots, K$ **do**
4:      Simulate an event-time $\tau_i$ for each rate $\lambda_i(\boldsymbol{x}_t, \boldsymbol{v}_t)$
5:      $i^* \leftarrow \operatorname{argmin}_i\{\tau_i\}$ and $t_k \leftarrow t_{k-1} + \tau_{i^*}$
6:      $\boldsymbol{x}_{t_k} \leftarrow \boldsymbol{x}_{t_{k-1}} + \tau_{i^*} \boldsymbol{v}_{t_{k-1}}$              ▷ *Update position*
7:      $\boldsymbol{v}_{t_k}^{i^*} \leftarrow -\boldsymbol{v}_{t_{k-1}}^{i^*}$                   ▷ *Update velocity*
8: **end for**
9: **Outputs:** Zig-Zag skeleton $\{t_k, \boldsymbol{x}_{t_k}, \boldsymbol{v}_{t_k}\}$.

---

# 3 Continuous tempering with a point mass

## 3.1 Continuous tempering

Continuous tempering is an approach to improve mixing of MCMC and related sampling algorithms. It introduces a second distribution $\pi_0$, called the *base* distribution, which is assumed to have a density $\pi_0(\boldsymbol{x}) = \frac{q_0(\boldsymbol{x})}{Z_0}$. The idea is that this density will be simple to simulate from. For any $\beta \in [0, 1]$ it is now possible to define a distribution which interpolates between $\pi$ and $\pi_0$ in that its density is $\pi(\boldsymbol{x})^\beta \pi_0(\boldsymbol{x})^{(1-\beta)}$. Continuous tempering then defines a joint distribution for $\beta$ and $\boldsymbol{x}$. This distribution has un-normalised density $q(\boldsymbol{x}, \beta)$, on $\mathbb{R}^d \times [0, 1]$, defined to be

$$q(\boldsymbol{x}, \beta) = q_0(\boldsymbol{x})^{1-\beta} q(\boldsymbol{x})^\beta.$$

In practice, tempering methods often introduce a pseudo prior, $p(\beta)$ on $\beta \in [0, 1]$, and sample from the distribution proportional to $p(\beta)q(\boldsymbol{x}, \beta)$. The idea of introducing the prior is that it can be tuned

to give a marginal distribution on $\beta$ that has reasonable mass across all of the interval $[0, 1]$. Without a prior, the distribution $q$ is likely to put almost all mass on $\beta$ close to 0 or 1.

By construction, conditional on satisfying $\beta = 1$, samples drawn from $q(\boldsymbol{x}, \beta)$ are distributed according to $\pi$. However, if we use a continuous prior, samples will almost-surely *not* lie in that set, and we have to use importance sampling to correct for $\beta \neq 1$. This leads to a weighted sample from $\pi$, with the Monte Carlo accuracy of the resulting approach depending greatly on the variability of the weights introduced by importance sampling.

To overcome this issue, we use a $p(\beta)$ that introduces a point mass at $\beta = 1$. Specifically, for $\alpha \in \mathbb{R}$, and an arbitrary (possibly non-normalised) probability density function $\kappa$ on the interval $[0, 1)$, we define

$$p(\beta) = (1 - \alpha)\kappa(\beta)\mathbf{1}_{(0 \leq \beta < 1)} + \alpha\kappa(\beta)\mathbf{1}_{(\beta=1)}.$$

With the above prior, we can define the augmented target

$$\omega(\mathrm{d}\boldsymbol{x}, \mathrm{d}\beta) \propto q(\boldsymbol{x}, \beta)(1 - \alpha)\kappa(\beta)\mathrm{d}\boldsymbol{x}\mathrm{d}\beta + q(\boldsymbol{x})\alpha\kappa(\beta)\mathrm{d}\boldsymbol{x}\delta_{\beta=1}. \tag{1}$$

We can extend the Zig-Zag process to sample from such a distribution. The next subsection describes the extension. The usual advantages of subsampling and transdimensional sampling using PDMPs apply for the tempered extension.

### 3.2 Continuously-tempered Zig-Zag

Unlike most MCMC samplers, Zig-Zag is a continuous-time process. This makes it simpler to deal with the point-mass at $\beta = 1$, using ideas from Chevallier et al. [2020]: we simulate paths for $\beta \in [0, 1)$ until $\beta$ hits the boundary at $\beta = 1$; when this happens, we set the velocity component for $\beta$, denoted as $\boldsymbol{v}_t^{d+1}$, to zero so that the Zig-Zag process explores the distribution $\omega$ restricted to $\beta = 1$; the Zig-Zag process stays with $\beta = 1$ until the first event in a new Poisson process, at which point we set the velocity component for $\beta$ to $-1$ to allow exploration of the distribution for $\beta \in [0, 1)$. We define the rate of the events when we leave $\beta = 1$ to be $\eta$. Algorithm 2 provides a sketch of the relevant modifications of the standard Zig-Zag process for events concerning the inverse temperature variable $\beta$.

---

**Algorithm 2** Tempered Zig-Zag algorithm

---

1: Run Zig-Zag for $\beta_t \in [0, 1)$ until first time $t$ at which trajectory hits $\beta = 1$
2: $\boldsymbol{v}_t^{d+1} \leftarrow 0$                                                    ▷ *Kill the velocity component for $\beta$*
3: Simulate an event-time $\tau$ with rate $\eta$
4: Run the Zig-Zag process with $\beta = 1$, for time $\tau$                              ▷ *Sample $\pi$*
5: $\boldsymbol{v}_{t+\tau}^{d+1} \leftarrow -1$                                           ▷ *Reintroduce $\beta$*
6: Goto Step 1.

---

An appropriate choice of rate to ensure that the resulting algorithm generates a process with $\omega$ as its limiting distribution is given in the following result (the proof of is located in Section 1 of the supplement).

**Theorem 1** *Assuming $\kappa(\beta)$ is continuous and the rate function in Algorithm 2 is chosen as*

$$\eta = \frac{1 - \alpha}{2\alpha},$$

*Then, the resulting Zig-Zag process has $\omega$ as a stationary distribution.*

The theorem statement above considers the case where $\kappa$ is continuous but we note in the proof that more general choice is possible with a slight adjustment to the rate.

Importantly, the rate $\eta(\boldsymbol{x})$ is constant, and thus simulating the time at which we transition from $\beta = 1$ to $\beta < 1$ is independent of the path of the process and simulating the event time is simple. While we have described the use of continuous tempering as an extension of the Zig-Zag process, the same construction readily extends to other PDMP samplers.

**Remark 1** *A tempered version of any PDMP-based sampler can be constructed more generally by adding a tempering variable $\beta$ with associated velocity $\pm 1$ and Zig-Zag rate function (irrespective of the base PDMP). When $\beta$ hits 1, the process reduces to the PDMP sampler on $\pi$ and is reintroduced with rate given by that of Theorem 1. See Remark 1 in the supplement for additional details.*

### 3.3 Importance sampling estimator

By construction, continuously-tempered Zig-Zag will spend a substantial amount of time in states with $\beta = 1$, and thus the states at those times can give draws from $\pi(\boldsymbol{x})$. We can use importance sampling ideas from Graham and Storkey [2017] to re-weight samples when $0 \le \beta < 1$ to give weighted samples from $\pi(\boldsymbol{x})$; but this only applies if $\kappa(\beta) \propto \xi^{1-\beta}$ for some constant $\xi$.

The idea of this approach is that, for $\beta \in [0, 1)$, we can marginalise out $\beta$ from $\omega$, and the marginal distribution for $\boldsymbol{x}$ is

$$\omega_{[0,1)}(\boldsymbol{x}) = \int_0^1 \kappa(\beta) q(\boldsymbol{x}) \left( \frac{q_0(\boldsymbol{x})}{q(\boldsymbol{x})} \right)^{1-\beta} \mathrm{d}\beta = q(\boldsymbol{x}) \int_0^1 \left( \frac{\xi q_0(\boldsymbol{x})}{q(\boldsymbol{x})} \right)^{1-\beta} \mathrm{d}\beta.$$

Defining $\Delta(\boldsymbol{x}) = \log q_0(\boldsymbol{x}) + \log \xi - \log q(\boldsymbol{x})$, the above is equal to

$$q(\boldsymbol{x}) \int_0^1 \exp\{(1-\beta)\Delta(\boldsymbol{x})\}\mathrm{d}\beta = \exp\{\Delta(\boldsymbol{x})\}\Delta(\boldsymbol{x})^{-1}(1 - \exp\{-\Delta(\boldsymbol{x})\}) = \frac{\exp\{\Delta(\boldsymbol{x})\} - 1}{\Delta(\boldsymbol{x})}.$$

Thus, if we have this as our proposal distribution, then the corresponding importance sampling weights will be $w(\boldsymbol{x}) = \Delta(\boldsymbol{x})/[\exp\{\Delta(\boldsymbol{x})\} - 1]$. However, this will involve additional post-sampling computation of the importance weights at the samples taken from the Zig-Zag trajectory.

### 3.4 Calibration of $\kappa(\beta)$

The efficiency of continuous tempering depends on an appropriate choice of $\kappa$ and $\alpha$. The choice of $\kappa$ also arises in path sampling Gelman and Meng [1998] and in simulated tempering Marinari and Parisi [1992]. A common approach from this literature is to choose $\kappa$ so the resulting marginal for $\beta$ is uniform. This choice is made to balance a non-negligible amount of time at $\beta = 1$, while simultaneously allowing occasional excursions to lower inverse temperatures to help mix, e.g. between modes. We consider how this choice for $\kappa$ may be adapted to our work considering the point mass at 1 and additional parameter $\alpha$.

For $\beta \in [0, 1]$ define, $Z(\beta) = \int_{\mathbb{R}^d} q(\boldsymbol{x}, \beta)d\boldsymbol{x}$. The induced $\beta$-marginal of $\omega$ for $\beta_0 \in [0, 1)$ is

$$\omega(\beta_0) = \frac{(1-\alpha)\kappa(\beta_0)Z(\beta_0)}{(1-\alpha)\int_0^1 \kappa(\beta)Z(\beta)d\beta + \alpha\kappa(1)Z(1)}.$$

The above suggests that an appropriate choice is $\kappa(\beta) \propto Z(\beta)^{-1}$, as this would both induce $\beta$ to be marginally *uniform* under $\omega$ for $\beta \in [0, 1)$, and cause the parameter $\alpha$ to directly represent the probability that $\beta = 1$ under $\omega$.

In our work, we choose $\kappa(\beta) \propto \widehat{Z}(\beta)^{-1}$, where $\widehat{Z}(\beta) = \exp\left\{ \sum_{k=0}^{m-1} a_k \beta^k \right\}$ for some coefficients $\{a_k\}_{k=0}^{m-1}$. The restriction to polynomial terms allows for simple implementation of thinning bounds following Sutton and Fearnhead [2021]. Since $Z(\beta)$ is unknown in practice, a common approach is to replace it with an approximation. Following ideas from the path sampling literature [Gelman and Meng, 1998, Yao et al., 2020], we can make use of numerical integration to form an estimate of $\log Z(\beta)$ using a pilot run of the sampler. The terms $\{a_k\}_{k=0}^{m-1}$ may be estimated based on this approximation (see Section 6.3 in the supplement for details).

## 4 Numerical experiments

### 4.1 Mixture of Gaussians

In our first example, the target corresponds to a mixture of Gaussian distributions with equal weights and variances so our target has unnormalised density

$$q(\boldsymbol{x}) = \sum_{i=1}^K \exp \left( -\frac{1}{2\sigma^2} (\boldsymbol{x} - \boldsymbol{\mu}_i)^\top (\boldsymbol{x} - \boldsymbol{\mu}_i) \right),$$

where $K = 5$, $\sigma^2 = 0.2$ and $\{\boldsymbol{\mu}_1, ..., \boldsymbol{\mu}_5\}$ were generated uniformly on the region $[0, 10] \times [0, 10]$ and are given in Table 1 of the supplement. Figure 1 plots the PDMP trajectories for the tempered and untempered Zig-Zag sampler, in addition to the trajectories for inverse temperature fixed at $\beta = 1$. For this example we choose $q_0(\boldsymbol{x})$ to be a Gaussian $\mathcal{N}(\nu, \Sigma)$ centred at $\nu = (5, 5)^T$ with covariance matrix $\Sigma = 2\mathbf{I}_2$.

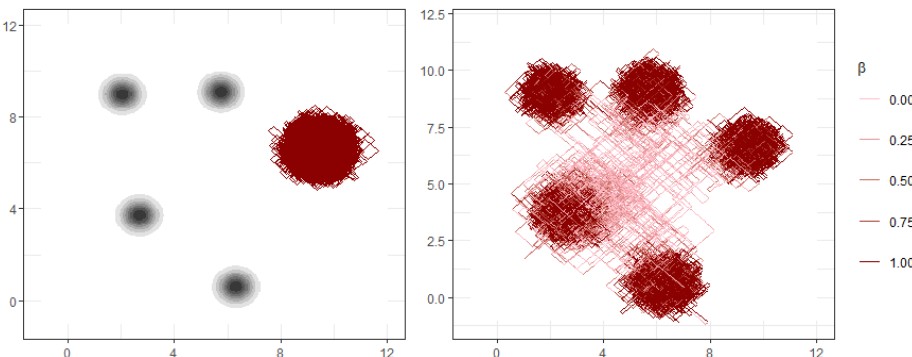

Figure 1: Trajectories of the Zig-Zag process simulated for 30,000 events for a multi-modal Gaussian mixture model, Zig-Zag (left) and continuously-tempered Zig-Zag (right) with $\alpha = 0.7$.

We define the *work normalised efficiency* as the square root of the total number of gradient (or target) evaluations multiplied by the Mean Square Error (MSE). This quantity includes the additional target evaluations required for the importance weights in Zig-Zag (IS). We consider the MSE for the first and second marginal moments of the Gaussian mixture model. Relative work normalised efficiency is then the ratio of this metric relative to a competitor. Table 1 shows the relative work normalised efficiency for the methods comparative to the standard (untempered) Zig-Zag sampler. A relative efficiency of 2 indicates the proposed method is twice as efficient as the standard Zig-Zag. All methods were simulated for 50,000 event-times, with the first 40% used as burnin in the standard Zig-Zag and used for both burn-in and estimating the polynomial $\kappa(\beta)$ in the tempered samplers.

Table 1: Work normalised efficiency for the first two moments of a Gaussian mixture relative to the standard Zig-Zag sampler (averaged over 20 replications).

| Method | $\alpha$ | $\omega(\beta = 1)$ | Relative work normalised efficiency | | | | Thinning efficiency |
|---|---|---|---|---|---|---|---|
| | | | $\mathbb{E}[X_1]$ | $\mathbb{E}[X_2]$ | $\mathbb{E}[X_1^2]$ | $\mathbb{E}[X_2^2]$ | |
| Zig-Zag | 1 | 1 | 1 | 1 | 1 | 1 | 0.057 |
| Zig-Zag CT | 0.800 | 0.789 | 4.641 | 4.366 | 4.568 | 5.216 | 0.080 |
| | 0.700 | 0.703 | 8.079 | 5.024 | 8.447 | 6.202 | 0.090 |
| | 0.500 | 0.499 | 10.963 | 7.134 | 10.265 | 9.047 | 0.114 |
| | 0.300 | 0.302 | 13.145 | 9.388 | **14.838** | 11.885 | 0.139 |
| | 0.200 | 0.197 | **14.153** | 9.479 | 13.235 | 11.225 | 0.153 |
| | 0.100 | 0.097 | 12.550 | **12.057** | 13.145 | **12.958** | 0.167 |
| Zig-Zag (IS) | 0 | 0 | 10.640 | 11.689 | 11.038 | 12.713 | **0.301** |

The standard Zig-Zag is not able to explore the multiple modes, yielding poor estimates of the first and second moments — see section 6.2 of the supplement for RMSE performance. We also compare to a direct Zig-Zag analogue of the continuously-tempered HMC algorithm [Graham and Storkey, 2017] where no mass is given to $\beta = 1$ and $\kappa(\beta) \propto 1$, the estimator is based on importance sampling as defined in Section 3.3. We find that the tuning procedure yielded precise control over the time spent at $\beta = 1$, and that for $\alpha$ between 0.1 and 0.3 the results are similar. Tempering with a point mass was found to give more efficient estimates of the first and second moments. This may be because the importance sampling estimate spends more time at $\beta \approx 0$. In addition to exhibiting better statistical performance, the tempered versions of the Zig-Zag sampler have better computational properties. The thinning efficiency, measured as the proportion of proposals that result in an event-time simulation is $\approx .06$ at $\beta = 1$, but improves significantly when the sampler can transition to lower temperatures.

While the importance sampling estimator for $\beta = 0$ has the best thinning efficiency, we note that it requires additional post processing to evaluate the importance sampling weights.

Table 2 provides further simulations to compare our tempering approach with both reversible [Woodard et al., 2009] and non-reversible [Syed et al., 2022] parallel tempering (denoted R-PT and NR-PT respectively). A Zig-Zag kernel is used at each temperature level and run for $S \in \{0.1, 1, 2\}$ units of stochastic time. We use a geometric temperature sequence $[1, a^1, a^2, ..., a^n]$ as commonly recommended in the literature [Tawn et al., 2020] and consider results for $a \in \{0.1, 0.3, 0.5, 0.7\}$, with $n \in \{3, 5, 7\}$. Results are again relative to standard (untempered) Zig-Zag in terms of work normalised efficiency. **We present the best performing parallel tempering result for each choice of** $a$. The full set of results is provided in Section 7 of the supplement. Our methods show similar performance to well-tuned NR-PT and outperform R-PT.

Table 2: Work normalised efficiency for the first two moments of a Gaussian mixture for parallel tempering relative to the standard Zig-Zag (averaged over 20 replications). Results are only shown for the best performing choice of $S$ and $n$ for each choice of $a \in \{0.1, 0.3, 0.5, 0.7\}$. Extended results may be found in the supplement.

| Method | $a$ | $n$ | $S$ | Relative work normalised efficiency | | | |
| --- | --- | --- | --- | --- | --- | --- | --- |
| | | | | $\mathbb{E}[X_1]$ | $\mathbb{E}[X_2]$ | $\mathbb{E}[X_1^2]$ | $\mathbb{E}[X_2^2]$ |
| NR-PT | 0.1 | 7 | 1 | 9.160 | 11.029 | 9.441 | 12.883 |
| NR-PT | 0.3 | 5 | 1 | 11.626 | **14.757** | 11.679 | **15.972** |
| NR-PT | 0.5 | 5 | 2 | 11.072 | 12.320 | 11.352 | 13.951 |
| NR-PT | 0.7 | 7 | 1 | 12.185 | 8.942 | **12.956** | 10.512 |
| R-PT | 0.1 | 5 | 1 | 7.142 | 6.750 | 7.382 | 7.448 |
| R-PT | 0.3 | 5 | 1 | 9.549 | 7.645 | 9.912 | 8.949 |
| R-PT | 0.5 | 3 | 1 | 10.136 | 7.672 | 10.830 | 8.842 |
| R-PT | 0.7 | 7 | 1 | **12.275** | 11.273 | 12.591 | 12.648 |

## 4.2 Transdimensional example

We apply the tempered Zig-Zag to the challenge of sampling a transdimensional distribution. Such target distributions arise naturally in variable selection problems. The resulting target (posterior) distribution is a discrete mixture of $2^p$ models, where $p$ is the number of variables in the dataset.

Such a setting produces a challenge, as typical samplers such as HMC do not extend naturally to such spaces. On the other hand, PDMPs have recently been extended to sample transdimensional distributions [Chevallier et al., 2020, 2021, Bierkens et al., 2021]. This extension employs within-model gradient information to efficiently explore the space and jumps between models when a parameter hits zero. The approach is beneficial as an informative likelihood function's gradient will direct less informative variables towards zero and more informative ones away from zero, aiding exploration of the sampler.

This example explores how tempering a target with a point mass can improve performance over the standard transdimensional Zig-Zag process. Specifically, we define a family of example problems of increasing difficulty in the sense that for higher values of an underlying parameter $m$, separation between the mode of the slab component and the spike at zero is increased. Allow

$$q(\boldsymbol{x}) = \prod_{i=1}^{2} (w\phi(x_i; m, \sigma^2) + (1-w)\delta_0(x_i)), \tag{2}$$

where $\phi(x_i; m, \sigma^2)$ is the normal density and $w = 0.5$ is the probability of a variable being included. We fix $\sigma^2 = 0.5$ and increase $m$ from $m = 0$ to $m = 4$. To enable tempering, define

$$q(\boldsymbol{x}, \beta) = \prod_{i=1}^{2} (w\phi(x_i; m\beta, \sigma^2) + (1-w)\delta_0(x_i)),$$

where at $\beta = 0$ the spike and slab is centred at zero, encouraging variables to enter and exit the model. While this is a somewhat artificial example, it is important to note that it encompasses precisely the

situations which transdimensional PDMPs will find challenging, and allows us to explore robustness for increasing levels of pathology in the purest setting possible.

Figure 2 (rightmost panel) displays the dynamics of the tempered Zig-Zag for this problem for the choice of $m = 4$. Note that the standard transdimensional Zig-Zag process (bottom left) becomes stuck in a single model whilst its tempered counterpart is able to transition easily between models and thus visit the required areas for which each coordinate is equal to zero.

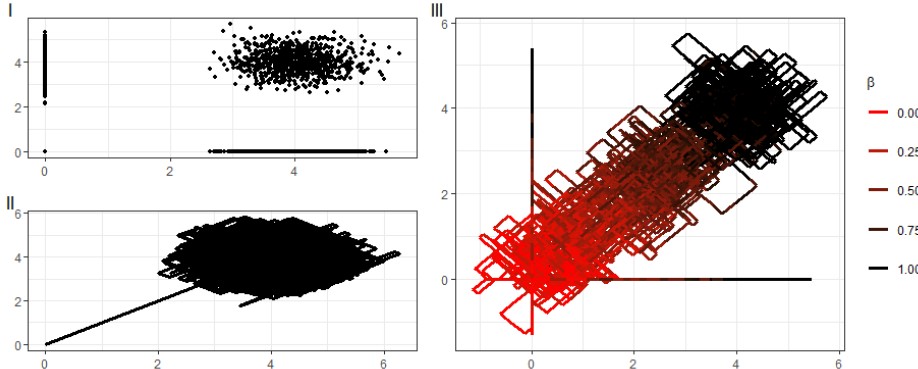

Figure 2: Sampling from (2) with $m = 4$. I. Exact samples from the spike-slab distribution. II. Standard Zig-Zag for $10^4$ event-times. III. Tempered Zig-Zag trajectories with $\alpha = 0.5$ for $10^4$ event-times.

Table 3: Absolute error of marginal mean and probability of inclusion for $X_1$ (averaged over 10 simulations for increasing $m$ in (2)).

|  |  | Mean Absolute Error | | | | |
| --- | --- | --- | --- | --- | --- | --- |
|  |  | $m = 0$ | $m = 1$ | $m = 2$ | $m = 3$ | $m = 4$ |
| $\mathbb{E}[X_1]$ | Zig-Zag | **0.005** | **0.018** | 0.633 | 1.498 | 1.998 |
|  | Zig-Zag CT | 0.007 | 0.025 | **0.022** | **0.047** | **0.214** |
| $\mathbb{P}(\|X_1\| > 0)$ | Zig-Zag | **0.009** | **0.012** | 0.322 | 0.500 | 0.500 |
|  | Zig-Zag CT | 0.010 | 0.023 | **0.008** | **0.018** | **0.055** |

Table 3 gives the absolute error for the Monte Carlo estimates of the marginal inclusion probabilities and marginal means. For lower $m$, the standard and tempered Zig-Zag perform similarly as the mode of the slab is close enough to ensure regular crossing of the zero-axis for the PDMP trajectories. However, as $m$ increases, the marginal probabilities of inclusion tend to 1 and the standard Zig-Zag process becomes stuck in a single model.

### 4.3 Boltzmann machine relaxation

Following Nemeth et al. [2019] and Graham and Storkey [2017], the final example considers sampling a continuous relaxation of a Boltzmann machine distribution. Full details surrounding the derivation of the distribution can be found in [Nemeth et al., 2019, Supplementary Material, Section D], though for the considered example the target density on $\mathbb{R}^{d_r}$ is of the form

$$q(\boldsymbol{x}) = \frac{2^{d_b}}{(2\pi)^{d_r/2} Z_b \exp(\frac{1}{2}\text{Tr}(\boldsymbol{D}))} \exp\left(-\frac{1}{2}\boldsymbol{x}^\top \boldsymbol{x}\right) \prod_{k=1}^{d_b} \cosh(\boldsymbol{q}_k^\top \boldsymbol{x} + b_k),$$

where $\boldsymbol{q}_k$ denotes the $k$-th row of $\boldsymbol{Q}$, which is a $d_b \times d_r$ matrix such that $\boldsymbol{Q}\boldsymbol{Q}^\top = \boldsymbol{W} + \boldsymbol{D}$, and $\boldsymbol{D}$ is an arbitrary diagonal matrix that ensures $\boldsymbol{W} + \boldsymbol{D}$ is positive semi-definite. The above is a continuous relaxation of the Boltzmann Machine distribution on $\{-1, 1\}^{d_b}$ with probability mass function

$$q(\mathbf{s}) = Z_b^{-1} \exp\left(\frac{1}{2}\boldsymbol{s}^\top \boldsymbol{W}\boldsymbol{s} + \boldsymbol{s}^\top \boldsymbol{b}\right).$$

The moments up to second order of the relaxation and the original (discrete) Boltzmann Machine distribution are related via $\mathbb{E}[\boldsymbol{X}] = \boldsymbol{Q}^\top \mathbb{E}[\boldsymbol{S}]$ and $\mathbb{E}[\boldsymbol{X}\boldsymbol{X}^\top] = \boldsymbol{Q}^\top \mathbb{E}[\boldsymbol{S}\boldsymbol{S}^\top] + \boldsymbol{I}$. We employ a similar experimental setup as in Nemeth et al. [2019], namely a 28-dimensional example which allows for exact computation via enumeration of the moments of $\boldsymbol{S}$, and hence in turn of $\boldsymbol{X}$ (the latter being useful for evaluating sampler performance). Again, we use work normalised efficiency taken relative to the standard (untempered) Zig-Zag as described in Section 4.1. The average mean square error for the first and second marginal moments over the 28 dimensions is used to evaluate the work normalised efficiency.

Table 4 displays the results. We find that the best performance is given by the Zig-Zag sampler with $\alpha = 0.2$ which outperforms the standard Zig-Zag ($\alpha = 1$) and the continuously-tempered importance sampling approach ($\alpha = 0$). For these experiments, the tuning of $\kappa$ is notably sub-optimal as the proportion of time spent at zero is not fully controlled by $\alpha$. Despite this, the samplers still offered substantial improvements over the importance sampling approach and the standard Zig-Zag. Further details on the experiment and specification of $\kappa$ may be found in Section 6.3 of the supplement. Additional tables recording the root mean square error of the methods may be found in Section 6.2 of the supplement.

Table 4: Average relative work normalised mean square error of the first and second marginal moments of the Boltzmann machine relaxation averaged over 20 simulations reported to 3 decimal places.

| Method | $\alpha$ | $\omega(\beta = 1)$ | Average relative work normalised efficiency | | Thinning efficiency |
|---|---|---|---|---|---|
| | | | $\mathbb{E}[X_k]$ | $\mathbb{E}[X_k^2]$ | |
| Zig-Zag | 1 | 1 | 1 | 1 | 0.146 |
| Zig-Zag CT | 0.700 | 0.632 | 2.025 | 1.065 | 0.187 |
| | 0.500 | 0.417 | 1.835 | 1.016 | 0.219 |
| | 0.300 | 0.231 | 2.021 | 0.795 | 0.251 |
| | 0.200 | 0.145 | **2.511** | **1.033** | 0.267 |
| | 0.100 | 0.076 | 1.332 | 0.549 | 0.284 |
| Zig-Zag CT (IS) | 0 | 0 | 1.253 | 0.390 | **0.505** |

We further compared our methods with reversible and non-reversible parallel tempering for a wide variety of tuning choices, and these results may be found in Section 7 of the supplement. We found that our methods appear to be less sensitive to tuning. However, for this example parallel tempering was able to achieve better performance than our continuously-tempered methods for certain choices of $a$, $S$ and $n$. Parallel tempering achieved an optimal average work normalised relative efficiency of 6.1 on the first marginal moment (NR-PT with $a = 0.5$, $S = 2$, $n = 7$) and 1.98 for the second marginal moment (R-PT with $a = 0.7$, $S = 2$, $n = 7$).

## 5  Discussion

We present a general strategy to improve the performance of PDMP samplers on challenging targets. The approach uses an extended distribution that uses an inverse temperature which interpolates between a challenging distribution of interest and a tractable base distribution. At lower values of $\beta$ the mixing will improve and we allow a point-mass at $\beta = 1$ to attain exact samples from the distribution. As a proof of concept, a numerical study of these ideas surrounding the Zig-Zag sampler was employed. This revealed that there are considerable benefits to the proposed extensions, as well as to introducing the point mass at $\beta = 1$. We compared our tempering with a point-mass extension of the Zig-Zag to a version based on continuously-tempered estimates that are given using importance sampling ideas from Graham and Storkey [2017]. The importance sampling approach requires restrictive choice of $\kappa$ and gave inferior performance. Our approach may be readily applied to other PDMP samplers and can incorporate assets of sampling with a PDMP such as sampling transdimensional spaces and using subsampling methods. Another avenue of research is in incorporating direct simulation from $q_0$ when the sampler hits $\beta = 0$ — such moves would be particularly useful when $q_0$ is a tractable multimodal distribution. Another avenue for research would be to consider parallel versions of the continuous tempered PDMP. Additional moves could be made to swap temperatures between different parallel PDMPs analogous to parallel tempering.

## Broader impact

Effective sampling methods are instrumental in many probabilistic modelling approaches that account for uncertainty quantification (e.g., Bayesian Machine Learning approaches). Thus, it is important to employ reliable methods in obtaining such estimates, especially if they are used for downstream tasks and decisions. The methodological innovations discussed herein aim to produce more reliable results in the context of PDMP samplers, which are a growing area of research. The proposed approaches are more suited to sampling challenging multi-modal distributions with PDMP-based samplers. As the methods are presented as a proof of concept, and do not feature as an implemented component within any deployed artificial intelligence system, there are no immediate broader consequences related to their potential failures.

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
