# OpenReview forum: "Continuously Tempered PDMP samplers"
_NeurIPS.cc/2022/Conference — NeurIPS 2022 Accept_

### Official Review · Reviewer_Lpcb · 2022-06-23

**Rating:** 8
**Confidence:** 4
**Soundness:** 4 excellent
**Presentation:** 4 excellent
**Contribution:** 3 good

**Summary:**

This article presents a principled way to do continuous tempering in PDMP samplers, with positive probability of sampling from the actual target distribution. It discusses its ergodicity, implementation and optimisation with respect to the tempering distribution. The method is illustrated on a series of challenging examples.

**Questions:**

1. I am rather uncertain how Section 3.4 on the calibration of $\kappa$ differs from Section 2.1. of Yao et al. [2020]. It seems to me that Section 3.4 is a direct special case of the latter. Can you explain how your method is different from this? Note that I understand this is not the main contribution of the article, but believe this kinship between the two methods should be highlighted in case I am not mistaken.

2. The use of $t$ and $s$ in Section 3 of the supplementary material is somewhat fuzzy: $t$ sometimes is a free parameter (or the current time), and sometimes denote the next exent. Clarifying the notation (for instance by calling the event time $\tau$ as per the main text) would be useful. Referring to the supplementary material at the time of introducing the Zig-zag sampler would be useful too.

3. I believe that the proof that $\omega$ is an invariant distribution for the given rate (Theorem 1) is correct. However, I believe this to follow more from Proposition 7 of [6] than directly from Theorem 2 and 3; a comment on the other PDMP sampler would also be useful. Moreover, for completeness, it may be useful in the supplementary material to insist that $\omega$ is the **unique** stationary distribution of the chain.



**Limitations:**

One of the main limitations of PDMP is the need to sample from a non-homogeneous Poisson process. This is however thoroughly discussed in the supplementary material.

**Strengths And Weaknesses:**

Strengths:
--------------
1. This article is well written, and beside its contributions, offers a good overview of the zig-zag sampler.
2. The heuristics of the method are clear, and its symbiosis with PDMP samplers made clear.
3. The method is theoretically grounded, and the article clearly describes the sampling mechanism (thinning) for each example.
4. The applicability of tempering for transdimensional sampling is particularly convincing.


Weaknesses:
------------------
1. Because the paper insists that the method applies to any PDMP sampler, a quick review of PDMP samplers in general (as in, a short summary of, e.g., Fernhead et al. [2018]), allowing the reader to confirm that it indeed generalises, would have been welcome in the supplementary material.

2. While the experimental validation is thorough, the reported metrics hardly allow to understand which sampler is best due to the fact we are lacking information on the run time of each method. This is probably the main issue with the article. Given that the RMSE of the MCMC estimates is roughly speaking $O(\frac{1}{\sqrt{ESS}})$, and that $ESS$ should increase in average linearly with the run time of the experiment (supposing that the average time to simulate an event is constant), I believe a better metric to report would be $\frac{RMSE}{\sqrt{T}}$, where $T$ is the time (e.g., in seconds) to get 50,000 samples from the chains, similarly, the absolute error should then be divided by the time taken (please adapt the $\sqrt{T}$ terms the way you deem best to match the super efficiency of PDMPs). Would this be feasible? Alternatively (and perhaps better), instead of giving yourself a number of iterations in the chain, you could give yourself a time budget for each chain (for example the average time taken by your experiments) and report the error achieved in this amount of time. I tend to believe this could marginally change the conclusions as it would take into account the thinning efficiency directly into the result.

3. The estimator of Section 3.3 --- which I believe is essentially a Rao-Blackwellisation of the proposed method for a specific choice of tempering prior --- does not seem to be empirically evaluated at all. Instead, a less performing version is used which almost surely does not sample from the target. It would have been interesting to also check if the "sacrifice" of using a poorly adapted tempering prior was compensated by being able to use the intermediate samples in the case when there is a point mass at $\beta = 1$. However, this point is less important than the point 2 above.

Some minor comments and typos:
---------------------------------------------
1. The reference style changes between the main paper and the supplementary, making it hard to jump from one to the other.
2. The citations in the main paper are breaking the text, and it seems like the authors perhaps used "\citet{}" from natbib everywhere. This probably should be corrected for readability. For example "piecewise deterministic processes (PDMPs) Davis [1993]" should probably read "piecewise deterministic processes (PDMPs) [Davis, 1993]" or similar depending on the chosen style.
3. The text alternates between of "piece-wise" and "piecewise"
4. "called a piecewise deterministic processes" -> "called piecewise deterministic processes"
5. "The idea is that allowing mixing across this set will improve mixing" -> probably meant "moving across this set"
6. "transitions from the continuous" -> "transitions between ..."
7. "for an exponential period of time": this sounds like it stays there for a long time, where as you approximately meant "exponentially distributed"
8. "This will involve additional post-sampling computation of the importance weights at the samples taken from the Zig-Zag trajectory." I believe this cost will probably be negligible compared to the cost of simulating from the Poisson process, so this sentence feels overly pessimistic to me.
9. In the supplementary, the box plots are not linked to the text.
10. "Zig-Zag $\alpha = 1$ and importance" -> "Zig-Zag ($\alpha = 1$) and importance"
11. "and we allow a pointmass at  $\beta = 1$" -> "and we allow a pointmass at $\beta = 1$"?

---

> ### Author Response · Authors · 2022-08-02
> **Response to reviewer Lpcb**
>
> We thank the reviewer for your time and thoughtful review.  Please find below our response to your questions
>
> 1. The approach in Section 3.4 is indeed the same as the approach in Yao et al., which actually has its origins in Section 5.2 of [1]. That said, a key implementation difference is to use polynomial functions in $\kappa$ (as opposed to splines or otherwise), as this makes constructing bounds for the process more straightforward. We've added a comment on this and also clarified the link to related work.
>
> 2. We have modified the supplementary material as suggested. As suggested, a reference to the supplementary material is also included.
>
> 3. Proposition 7 of [6] is indeed related to the ZigZag sampler, albeit in a slightly different setting. In that paper, the target density has *discontinuities*. In our paper, the target density has a *Dirac* (atomic) component. Handling discontinuities is arguably more complicated than handling atoms for the ZigZag sampler, which is why Proposition 7 is required in [6]. To handle discontinuities, the trajectory should "reflect" on the normal of the discontinuity, which is not straightforward to do when the velocity space is discrete. By comparison, handling atomic components in the target requires only an "absorption" mechanism, and hence follows more directly from Theorem 2 and 3 of the cited work. Regarding uniqueness of stationary distribution: Theorem 1 was mistakenly stated in terms of ergodicity instead of invariance --- we have changed the result accordingly, and thus do not consider uniqueness which would require additional technical assumptions on the target.
>
> Response to Minor Comments:
>
> - We thank the reviewer for pointing out several typographical errors and inconsistencies, which have all been fixed in the new version.
>
> - Regarding the computational cost of additional post-sampling computation of importance weights, in general this is not as negligible as it may first appear. The reason for this is that one needs to choose a uniform discretisation of the simulated Zig-Zag trajectory, and then evaluate the target density at each point to construct the importance weights. In contrast, taking only the trajectory at inverse temperature one with the proposed approach does not require such computation, and one can even exploit the entire continuous path by analytically integrating simple functions (e.g., first moments).
>
> Response to Weaknesses:
>
> - Regarding possible generalization(s) of Theorem 1: We have included remark 1 in the main text and an additional remark in the supplementary material sketching the required modifications to construct tempered versions of other PDMP samplers. By using Zig-Zag dynamics for only the tempering variable the rate $\eta$ remains the same.
>
> - Regarding reporting RMSE scaled by computational effort. This comment was also brought to our attention by reviewer 2 so we summarise our response here: The supplement now reports (an appropriately normalized) MSE number of gradient or target evaluations.  We highlight that the overall conclusions do not change. In our experiments, we simulated all of the PDMP methods for the same number of event times, which means that they require an equivalent computational effort under the assumption that (excepting the importance sampling approach, which requires additional evaluations as part of post-processing) a perfect bound is used to simulate the process (i.e., one always accepts). While the above is an idealized scenario, the construction of efficient bounds for PDMP samplers is a topic of ongoing research, and work-normalized performance measures may change depending on the use of different bounds. Thus, we believe that reporting the RMSE results across methods for the same number of simulated events is a better indicator of a PDMP approach's longer-term potential.
>
> - Regarding using intermediate samples with a "poorly adapted tempering prior" - We had indeed made a similar observation to this point in the earlier stages of this work, and had experimented with using the choice of the "poorly adapted tempering prior'' and combining the IS estimator with the regular PDMP estimator in a number of ways (including using the difference of the two as a control variate for the latter). This was met with limited success. The conclusion was that the sacrifice of choosing $\kappa$ such that IS is tractable is not worthwhile. In general, such an approach is unreliable as it does not allow precise control of exactly how much and at what temperatures "exploration'' occurs, thus introducing additional complications. We also found that the problem was made worse by higher-dimensional settings (see figure below).
>
> [1] A. Gelman and X.-L. Meng. Simulating normalizing constants: From importance sampling to bridge
> sampling to path sampling (1998). Statistical science: a review journal of the Institute of Mathematical164
> Statistics, 13(2):163–185

---

> > ### Comment · Reviewer_Lpcb · 2022-08-03
> > **Acknowledgement of response**
> >
> > Thank you for your clarifications on the proofs, related works, and overall choice of methodology.
> >
> > I have read other reviewers comments, and overall I still believe that the paper is a strong contribution. This work shows that something that was not possible with classical MCMC samplers is now possible with PDMPs. This is ground enough for publication in my opinion.
> >
> > My common concern with Reviewer 77P1 has been solved (regarding the fact that extension to other PDMPs wasn't clear). I have also left a comment under of Reviewer Lpcb's review. Their request for a comparison with standard baselines seems grounded to me, although I do not think this is critical for publication of this work in its current state.
> >
> > Based on this, I have increase my overall score to 8, and the soundness assessment to 4.

---

### Official Review · Reviewer_rki5 · 2022-07-02

**Rating:** 6
**Confidence:** 5
**Soundness:** 4 excellent
**Presentation:** 3 good
**Contribution:** 2 fair

**Summary:**

The manuscript proposes to use a modification of the ZigZag sampler (or any almost any other PDMP sampler) to sample from complex (eg. multimodal) target distributions. The approach proceeds by introducing a "temperature" parameter $\beta \in [0,1]$ and an extended distribution with a Dirac mass at $\beta=1$ (corresponding to the target distribution of interest) and run a slightly-modified ZigZag sampler on the extended target. Samples obtained at temperatures $\beta < 1$ can still be used by using an importance sampling scheme, if necessary.

**Questions:**

1. Choosing the distribution $p(\beta)$ is indeed crucial in not-entirely-trivial situations. I think that this section should be expanding greatly. Choosing $\kappa(\beta) \propto 1/Z(\beta)$ is indeed a good strategy, I think, but quite hard to implement in practice in difficult situations. How do the authors think this should be done in practice? There are about 3 lines describing this and one would expect a much more thorough discussion since it is likely to have a very important impact on the method.

2. Implementing the Zig-Zag, and many other PDMP sampler, is not trivial (eg. simulating the event time, etc..) A metric of the type (accuracy)/(compute time) should be reported. For example, (accuracy) / (number of target or gradient-of-target evaluations) seems like an appropriate metric.

3. It is very standard and straightforward to use SMC methods, or parallel-tempering schemes, to tackle multi-modal situations. Furthermore, it is not that difficult to (almost) automatically tune these methods. I think that it is crucial to compare the proposed method to these standard alternatives (and report ESS or MSE per compute time!). Since the method is relatively simple (which is good), it seems natural to expect very thorough numerical experiments.

**minor remarks:**
1. Are we assuming $\kappa(\beta)=1$ so that $\int_{[0,1]} p(\beta) d\beta=1$?


**Limitations:**

NaN

**Strengths And Weaknesses:**

**Strength:**

The text is well-written and very easy to follow. Furthermore, the approach is quite natural and, to the best of my knowledge, novel.

**Weaknesses:**
The method is simple to implement (which is an advantage) and its working principles are intuitive. Unfortunately, I have not found the numerical experiments extremely convincing. As a matter of fact, after reading the paper, I am not concince that I would like to give the method a try (even if it is simple to implement). See next section for more detailed remarks. As of now, I am leaning towards a "weak reject" since I am finding the empirical evaluation section to not be up to the standard of NeuRIPS.

---

> ### Author Response · Authors · 2022-08-02
> **Response to rki5**
>
> We thank you for your careful review. Please find below our responses to your specific questions.
>
> Questions:
> 1.  Regarding recommendations for choosing $\kappa(\beta)$ so to be approximately proportional to $Z(\beta)^{-1}$, we have provided a more thorough discussion of this in Section 3.4 and the supplementary material. A similar motivation for setting $\kappa(\beta)\propto 1/Z(\beta)$ for $p(\beta)$ arises in path sampling and simulated tempering methods more generally. Our revised manuscript clarifies that our approach uses the same choice and notes that in doing so we are able to control the amount of time spent at $\beta=1$. We note that in practice $Z(\beta)$ is replaced with an approximation and an approximation based on path sampling identities may be used to estimate it. The specific form of the path sampling identity is given in the supplementary material (Section 6.2). As this problem has existed for related methodology of path sampling and simulated tempering, so alternative methods to estimate the form of $Z(\beta)$ may be used -- e.g. adaptive approaches or other methods as described in Section 5.1 of [1].
>
> 2. The supplement now reports (an appropriately normalized) MSE$\times$ number of gradient or target evaluations for Example 1 and 3 (a table reporting results for Example 1 is included below). Example 2 is already work normalised since all thinning proposals are accepted for this example. The numbers reported in Table 1 are reported relative to the standard Zig-Zag sampler. So a value of 2 would is interpreted as performing twice as well as the standard Zig-Zag.
>
> ***Table 1: Work normalised Gaussian Mixture model***
> |method    | alpha| prob_1|   Ex.1|   Ex.2|  Ex2.1|  Ex2.2|
> |:---------|-----:|------:|------:|------:|------:|------:|
> |Zig-Zag   |   1.0|  1.000|  1.000|  1.000|  1.000|  1.000|
> |Zig-Zag CT  |   0.8|  0.789|  4.641|  4.366|  4.568|  5.216|
> |  |   0.5|  0.499| 10.963|  7.134| 10.265|  9.047|
> |  |   0.3|  0.302| 13.145|  9.388| **14.838**| 11.885|
> |  |   0.2|  0.197| **14.153**|  9.479| 13.235| 11.225|
> |  |   0.1|  0.097| 12.550| **12.057**| 13.145| **12.958**|
> |Zig-Zag CT (IS) |   0.0|  0.000| 10.640| 11.689| 11.038| 12.713|
>
> We highlight that the overall conclusions do not change. We retain the original tables in the main paper for the following reason: In our experiments, we simulated all PDMP methods for the same number of event times, which means that they require an equivalent computational effort under the assumption that (excepting the importance sampling approach, which requires additional evaluations as part of post-processing) a perfect bound is used to simulate the process.  While the above is an idealized scenario, the construction of efficient bounds for PDMP samplers is a topic of ongoing research, and work-normalized performance measures may change depending on the use of different bounds. Thus, we believe that reporting the RMSE results across methods for the same number of simulated events is a better indicator of a PDMP approach's longer-term potential.
>
> 3. SMC samplers (with temperature annealing) and parallel tempering are indeed standard approaches for multi-modal situations, and exploit a similar fundamental idea: improve mixing by targeting an augmented space which admits the original target as some appropriate marginal. The experiments in the paper designed with primary contribution of the work in mind: to illustrate the benefits that follow from applying such ideas to PDMP. With respect to that goal, we believe the experiments are sufficient in making the benefits of the approach clear.  We did not provide comparisons to other methods for the simple reasons that (i) we do not make claims such as being able to achieve e.g., state-of-the-art MCMC sampling performance for multimodal settings across all possible methods, but rather provide a natural strategy to improve performance in such settings in the context of PDMP samplers, and (ii) because there are a large number of choices inherent to SMC and parallel tempering (PT) methods, which results in somewhat of an "apples to oranges" when it comes to making precise comparisons to those methods. In light of such considerations, we opted to keep the experiments straightforward and simple, so not to detract from illustrating the primary contribution.
>
> Regarding Minor Remark 1: It is not required that $\kappa(\beta) = 1$, or more generally that $p(\beta)$ be appropriately normalized. The reason for this is that the sampler does not require the augmented target to be properly normalized, and similarly the value of $\kappa(1)$ appears only as a constant in the conditional distribution when $\beta=1$.
>
> [1] A. Gelman and X.-L. Meng. Simulating normalizing constants: From importance sampling to bridge
> sampling to path sampling (1998). Statistical science: a review journal of the Institute of Mathematical
> Statistics, 13(2):163–185

---

> > ### Comment · Reviewer_rki5 · 2022-08-03
> > **still expecting some comparisons with standard baselines**
> >
> > Thank you for the clarifications.
> > 1. noted on the expanded discussions of the tuning of the auxiliary distribution (although it would have been good to reproduce some of the expanded discussions here).
> > 2. noted that the experiments reported in the main text are already more or less normalized for computational cost.
> > 3. I entirely agree with the authors that one cannot indeed expect "state-of-the-art MCMC sampling performance for multimodal settings across all possible methods. Still, I believe that the audience of NeurRIPS (I would certainly do) would still be expecting some sort of comparisons with very standard methods such as SMC or parallel tempering. Indeed, there are a few things to take into account (eg. what Markov kernel to use), but there are a few very standard choices (MALA or HMC proposals). And it is very easy to automate the choice of temperature. The coment "we opted to keep the experiments straightforward and simple" feels a bit weak to me -- it **is** very simple to implement these baselines. As commented above, it is somehow fine whether the proposed method is not **uniformly** beating this very strong baseline -- still, NeurIPS readers are expecting this type of comparisons to be performed. At the end of the day, if the proposed method is **uniformly** worse than these standard baselines, it **is** a bit worrying.
> >
> > About the minor remark, noted that $p(\beta)$ is not assumed normalized. I may suggest to maybe slightly edit equation line (128). Maybe use \propto then?
> >
> > As of now, I still feel a bit uncomfortable suggesting "acceptance" since there is no comparison with very standard baselines.

---

> > > ### Comment · Reviewer_Lpcb · 2022-08-03
> > > **Comment on use of baselines**
> > >
> > > After reading Reviewer rki5's review, I tend to agree that a comparison to at least parallel tempering (e.g., https://proceedings.mlr.press/v139/syed21a.html, which is non-reversible and works on optimised tempering schedules too) would be a nice and fairly easy comparison, even though I disagree on the criticality of doing so and still think this work will have a strong impact without it. An interesting statistics to report would have been the "average number of events necessary to go from $\beta = 0$ to $\beta=1$".
> > >
> > > On the other hand, I can also understand the point of view of the authors: parallel tempering and tempered SMC require running many parallel chains, which is not necessarily compatible with the "one-CPU" framework of the current work and may feel a bit out of place.
> > > In fact, I do not believe that comparing with tempered SMC methods is as easy as suggested by Reviewer rki5: to the best of my knownledge, SMC sampling for transdimensional sampling should be done using *interacting* SMC samplers in one form or another https://www.sciencedirect.com/science/article/abs/pii/S0167947307003398.

---

> > > > ### Comment · Reviewer_rki5 · 2022-08-03
> > > > **comments on baselines**
> > > >
> > > > 1. running Simulated Tempering does seem very easy to do: alternate (1) use PDMP to explore a fixed temperature for some amount of time T>0 (2) attempt a change of temperature [as in usual Simulated Temperaing]. This seems like a very natural thing to look at, at least to motivate the usefulness of running PDMP on joint space (state,temperature)
> > > > 2. leaving the transdimensional example aside (I do not know well this type of problems), running parallel/simulated tempering, SMC, etc.. is entirely straightforward. And running everything on a single CPU is the rule rather than the exception
> > > > 3. I believe that it is a tradition at NeurIPS (and similar conferences) to carefully compare with baselines approaches. Again, it is indeed not necessary (and probably not possible) to beat these baselines in all and every situations. But gaining some intuitions as to when one can expect a newly developped method to be competitive is important.

---

> > > > > ### Author Response · Authors · 2022-08-08
> > > > > **Adding Baselines**
> > > > >
> > > > > We have conducted additional simulations to compare our tempering approach with both reversible [1] and non-reversible [2] parallel tempering (PT) approaches. A Zig-Zag kernel is used at each temperature level and run for S= 0.1, 1, 2 units of stochastic time on each tempering level. We use a geometric temperature sequence [1, a^1, a^2, ..., a^n] as commonly recommended in the literature, eg [3], and consider results for a=0.1,0.3,0.5,0.7, with n=3,5,7. Results are relative to untempered Zig-Zag in terms of work normalised efficiency (higher is better). Our approach beats [R] PT for the tunings considered and gives similar performance to [NR] PT. *Some simulations are still running on the cluster* and will be added to the final supplement (and here if complete by closing). A selection of these results will be included in the final paper.
> > > > >
> > > > > [1] Woodard, et al. "Sufficient Conditions for Torpid Mixing of Parallel and Simulated Tempering." EJP 2009.
> > > > >
> > > > > [2] Syed et al "Non-reversible parallel tempering: A scalable highly parallel MCMC scheme". JRSSB 2022
> > > > >
> > > > > [3] Tawn et al "Optimal Temperature Spacing for Regionally Weight-preserving Tempering" 2018
> > > > >
> > > > > We use PT1 in [1] for reversible PT and the switching kernel of [2] for non-reversible PT

---

> > > > > > ### Author Response · Authors · 2022-08-08
> > > > > > **Gaussian Mixture Model**
> > > > > >
> > > > > > ZigZag for reference:
> > > > > >
> > > > > > |method        | alpha| prob_1|  Ex.1|  Ex.2| Ex2.1| Ex2.2|
> > > > > > |:-------------|-----:|------:|-----:|-----:|-----:|-----:|
> > > > > > |Zig-Zag       |   1.0|    1.0|  1.00|  1.00|  1.00|  1.00|
> > > > > > |Zig-Zag CT    |   0.3|    0.3| 13.15|  9.39|**14.84**| 11.89|
> > > > > > |              |   0.2|    0.2|**14.15**|  9.48| 13.24| 11.23|
> > > > > > |              |   0.1|    0.1| 12.55|**12.06**| 13.14|**12.96**|
> > > > > > |              |   0.0|    0.0| 10.64| 11.69| 11.04| 12.71|
> > > > > >
> > > > > > |method   (a)  | n    | S     |  Ex.1|  Ex.2| Ex2.1| Ex2.2|
> > > > > > |:-------------|-----:|------:|-----:|-----:|-----:|-----:|
> > > > > > |[NR] PT (0.1) |   3.0|    0.1|  4.55|  5.28|  4.92|  6.02|
> > > > > > |[NR] PT (0.1) |   5.0|    0.1|  7.27|  6.31|  7.38|  7.43|
> > > > > > |[NR] PT (0.1) |   7.0|    0.1|  7.20|  5.82|  8.14|  6.71|
> > > > > > |[NR] PT (0.1) |   3.0|    1.0|  8.32|  8.48|  8.69| 10.02|
> > > > > > |[NR] PT (0.1) |   5.0|    1.0|  6.64|  5.69|  6.90|  6.78|
> > > > > > |[NR] PT (0.1) |   7.0|    1.0|  9.16| 11.03|  9.44| 12.88|
> > > > > > |[NR] PT (0.1) |   3.0|    2.0|  7.08|  8.21|  7.68|  9.26|
> > > > > > |[NR] PT (0.1) |   5.0|    2.0|  9.25|  7.33|  9.31|  8.81|
> > > > > > |[NR] PT (0.1) |   7.0|    2.0|  7.17|  6.25|  7.51|  6.83|
> > > > > > |[NR] PT (0.3) |   3.0|    0.1|  6.54|  7.98|  7.19|  8.83|
> > > > > > |[NR] PT (0.3) |   5.0|    0.1|  7.79|  6.44|  7.74|  7.73|
> > > > > > |[NR] PT (0.3) |   7.0|    0.1|  7.89|  7.95|  8.25|  9.38|
> > > > > > |[NR] PT (0.3) |   3.0|    1.0| 11.02|  8.65| 12.54|  9.72|
> > > > > > |[NR] PT (0.3) |   5.0|    1.0| 11.63|**14.76**| 11.68|**15.97**|
> > > > > > |[NR] PT (0.3) |   7.0|    1.0| 10.53| 11.78| 10.31| 12.31|
> > > > > > |[NR] PT (0.3) |   3.0|    2.0| 10.77|  9.29| 11.29| 11.11|
> > > > > > |[NR] PT (0.3) |   5.0|    2.0|  9.20| 11.16|  9.60| 13.85|
> > > > > > |[NR] PT (0.3) |   7.0|    2.0| 11.07| 10.91| 11.45| 12.64|
> > > > > > |[NR] PT (0.5) |   3.0|    0.1|  7.83|  4.43|  7.99|  5.30|
> > > > > > |[NR] PT (0.5) |   5.0|    0.1|  6.23|  7.58|  6.47|  8.01|
> > > > > > |[NR] PT (0.5) |   7.0|    0.1|  7.20|  6.45|  8.19|  7.47|
> > > > > > |[NR] PT (0.5) |   3.0|    1.0|  7.65|  8.54|  7.59| 10.57|
> > > > > > |[NR] PT (0.5) |   5.0|    1.0|  9.46| 11.16| 10.84| 12.37|
> > > > > > |[NR] PT (0.5) |   7.0|    1.0|**13.00**|  9.56|**13.68**| 11.43|
> > > > > > |[NR] PT (0.5) |   3.0|    2.0| 10.80|  7.78| 11.18|  9.18|
> > > > > > |[NR] PT (0.5) |   5.0|    2.0| 11.07| 12.32| 11.35| 13.95|
> > > > > > |[NR] PT (0.5) |   7.0|    2.0|  8.76|  8.53|  9.73|  9.28|
> > > > > > |[NR] PT (0.7) |   3.0|    0.1|  2.34|  1.24|  2.47|  1.37|
> > > > > > |[NR] PT (0.7) |   5.0|    0.1|  3.26|  4.01|  3.20|  4.68|
> > > > > > |[NR] PT (0.7) |   7.0|    0.1|  8.70|  4.79|  9.30|  5.65|
> > > > > > |[NR] PT (0.7) |   3.0|    1.0|  2.59|  1.64|  2.63|  1.87|
> > > > > > |[NR] PT (0.7) |   5.0|    1.0|  6.31|  5.96|  6.50|  6.75|
> > > > > > |[NR] PT (0.7) |   7.0|    1.0| 12.19|  8.94| 12.96| 10.51|
> > > > > > |[NR] PT (0.7) |   3.0|    2.0|  3.20|  1.51|  2.90|  1.80|
> > > > > > |[NR] PT (0.7) |   5.0|    2.0|  7.26|  7.02|  7.65|  8.01|
> > > > > > |[NR] PT (0.7) |   7.0|    2.0| 10.79|  8.23| 11.10|  9.00|
> > > > > > |[R] PT (0.1)  |   3.0|    0.1|  5.19|  4.73|  5.60|  5.36|
> > > > > > |[R] PT (0.1)  |   5.0|    0.1|  6.68|  5.14|  6.82|  6.10|
> > > > > > |[R] PT (0.1)  |   7.0|    0.1|  5.36|  4.84|  5.53|  5.80|
> > > > > > |[R] PT (0.1)  |   3.0|    1.0|  5.94|  6.08|  5.89|  7.53|
> > > > > > |[R] PT (0.1)  |   5.0|    1.0|  7.14|  6.75|  7.38|  7.45|
> > > > > > |[R] PT (0.1)  |   7.0|    1.0|  6.04|  5.81|  6.50|  6.55|
> > > > > > |[R] PT (0.1)  |   3.0|    2.0|  6.73|  5.72|  6.71|  6.66|
> > > > > > |[R] PT (0.1)  |   5.0|    2.0|  5.05|  5.90|  5.28|  6.53|
> > > > > > |[R] PT (0.1)  |   7.0|    2.0|  5.45|  5.62|  5.66|  6.30|
> > > > > > |[R] PT (0.3)  |   3.0|    0.1|  5.04|  6.37|  5.39|  7.65|
> > > > > > |[R] PT (0.3)  |   5.0|    0.1|  5.87|  7.24|  6.31|  8.07|
> > > > > > |[R] PT (0.3)  |   7.0|    0.1|  6.92|  6.39|  7.25|  7.89|
> > > > > > |[R] PT (0.3)  |   3.0|    1.0|  8.89|  7.11|  9.70|  8.12|
> > > > > > |[R] PT (0.3)  |   5.0|    1.0|  9.55|  7.64|  9.91|  8.95|
> > > > > > |[R] PT (0.3)  |   7.0|    1.0|  8.34|  7.74|  8.49|  9.52|
> > > > > > |[R] PT (0.3)  |   3.0|    2.0|  8.41|  6.95|  9.05|  7.91|
> > > > > > |[R] PT (0.3)  |   5.0|    2.0|  5.99|  5.91|  6.07|  6.91|
> > > > > > |[R] PT (0.3)  |   7.0|    2.0|  7.86|  5.31|  7.61|  6.86|
> > > > > > |[R] PT (0.5)  |   3.0|    0.1|  6.41|  4.74|  6.40|  5.31|
> > > > > > |[R] PT (0.5)  |   5.0|    0.1|  7.72|  7.05|  8.22|  8.16|
> > > > > > |[R] PT (0.5)  |   7.0|    0.1|  5.90|  9.59|  5.92| 10.64|
> > > > > > |[R] PT (0.5)  |   3.0|    1.0| 10.14|  7.67| 10.83|  8.84|
> > > > > > |[R] PT (0.5)  |   5.0|    1.0|  8.99|  9.41|  9.05|  9.82|
> > > > > > |[R] PT (0.5)  |   7.0|    1.0|  8.29|  8.16|  7.92|  9.89|
> > > > > > |[R] PT (0.5)  |   3.0|    2.0|  8.11|  6.15|  8.37|  6.87|
> > > > > > |[R] PT (0.5)  |   5.0|    2.0|  9.74|  8.32| 10.37|  9.12|
> > > > > > |[R] PT (0.5)  |   7.0|    2.0|  8.66|  7.23|  8.79|  9.18|
> > > > > > |[R] PT (0.7)  |   3.0|    0.1|  3.06|  1.17|  2.82|  1.39|
> > > > > > |[R] PT (0.7)  |   5.0|    0.1|  4.44|  2.95|  4.43|  3.59|
> > > > > > |[R] PT (0.7)  |   7.0|    0.1|  6.60|  4.92|  6.34|  6.12|
> > > > > > |[R] PT (0.7)  |   3.0|    1.0|  2.61|  1.72|  2.46|  2.04|
> > > > > > |[R] PT (0.7)  |   5.0|    1.0|  8.04|  6.06|  8.07|  7.09|
> > > > > > |[R] PT (0.7)  |   7.0|    1.0| 12.27| 11.27| 12.59| 12.65|
> > > > > > |[R] PT (0.7)  |   3.0|    2.0|  3.13|  1.76|  3.10|  2.02|
> > > > > > |[R] PT (0.7)  |   5.0|    2.0|  6.55|  6.10|  6.47|  7.10|
> > > > > > |[R] PT (0.7)  |   7.0|    2.0|  8.89|  7.93|  9.02|  9.58|

---

> > > > > > > ### Author Response · Authors · 2022-08-08
> > > > > > > **Boltzmann Machine**
> > > > > > >
> > > > > > > ZigZag for reference
> > > > > > > |method        | alpha| prob_1|      Ex |      Ex^2|
> > > > > > > |:-------------|-----:|------:|--------:|---------:|
> > > > > > > |Zig-Zag       |   1.0|   1.00|     1.00|      1.00|
> > > > > > > |Zig-Zag CT    |   0.7|   0.63|     2.03|      1.07|
> > > > > > > |     |   0.5|   0.42|     1.83|      1.02|
> > > > > > > |     |   0.3|   0.23|     2.02|      0.80|
> > > > > > > |     |   0.2|   0.14|   **2.51**|   **1.03**|
> > > > > > > |     |   0.1|   0.08|     1.33|      0.55|
> > > > > > > |     |   0.0|   0.00|     1.25|      0.39|
> > > > > > >
> > > > > > > |method   (a)  | n    | S     |      Ex |      Ex^2|
> > > > > > > |:-------------|-----:|------:|--------:|---------:|
> > > > > > > |[NR] PT (0.1) |   3.0|   0.10|     0.26|      0.09|
> > > > > > > |[NR] PT (0.1) |   5.0|   0.10|     0.29|      0.09|
> > > > > > > |[NR] PT (0.1) |   7.0|   0.10|     0.27|      0.08|
> > > > > > > |[NR] PT (0.1) |   3.0|   1.00|     1.73|      0.59|
> > > > > > > |[NR] PT (0.1) |   5.0|   1.00|     1.97|      0.58|
> > > > > > > |[NR] PT (0.1) |   7.0|   1.00|     2.18|      0.72|
> > > > > > > |[NR] PT (0.1) |   3.0|   2.00|     2.86|      0.84|
> > > > > > > |[NR] PT (0.1) |   5.0|   2.00|     3.22|      0.76|
> > > > > > > |[NR] PT (0.1) |   7.0|   2.00|     3.38|      1.05|
> > > > > > > |[NR] PT (0.3) |   3.0|   0.10|     0.31|      0.10|
> > > > > > > |[NR] PT (0.3) |   5.0|   0.10|     0.45|      0.15|
> > > > > > > |[NR] PT (0.3) |   7.0|   0.10|     0.46|      0.14|
> > > > > > > |[NR] PT (0.3) |   3.0|   1.00|     2.08|      0.68|
> > > > > > > |[NR] PT (0.3) |   5.0|   1.00|     2.58|      0.84|
> > > > > > > |[NR] PT (0.3) |   7.0|   1.00|     2.79|      0.78|
> > > > > > > |[NR] PT (0.3) |   3.0|   2.00|     3.59|      1.05|
> > > > > > > |[NR] PT (0.3) |   5.0|   2.00|     4.17|      1.34|
> > > > > > > |[NR] PT (0.3) |   7.0|   2.00|     4.45|      1.26|
> > > > > > > |[NR] PT (0.5) |   3.0|   0.10|     0.24|      0.15|
> > > > > > > |[NR] PT (0.5) |   5.0|   0.10|     0.44|      0.14|
> > > > > > > |[NR] PT (0.5) |   7.0|   0.10|     0.60|      0.20|
> > > > > > > |[NR] PT (0.5) |   3.0|   1.00|     0.96|      0.78|
> > > > > > > |[NR] PT (0.5) |   5.0|   1.00|     3.26|      1.03|
> > > > > > > |[NR] PT (0.5) |   7.0|   1.00|     3.22|      0.89|
> > > > > > > |[NR] PT (0.5) |   3.0|   2.00|     1.72|      1.30|
> > > > > > > |[NR] PT (0.5) |   5.0|   2.00|     5.63|      1.45|
> > > > > > > |[NR] PT (0.5) |   7.0|   2.00| **6.10**|      1.43|
> > > > > > > |[NR] PT (0.7) |   3.0|   0.10|     0.25|      0.15|
> > > > > > > |[NR] PT (0.7) |   5.0|   0.10|     0.30|      0.27|
> > > > > > > |[NR] PT (0.7) |   7.0|   0.10|     0.38|      0.28|
> > > > > > > |[NR] PT (0.7) |   3.0|   1.00|     0.72|      0.66|
> > > > > > > |[NR] PT (0.7) |   5.0|   1.00|     0.96|      1.08|
> > > > > > > |[NR] PT (0.7) |   7.0|   1.00|     2.49|      1.13|
> > > > > > > |[NR] PT (0.7) |   3.0|   2.00|     1.13|      1.27|
> > > > > > > |[NR] PT (0.7) |   5.0|   2.00|     1.37|      1.61|
> > > > > > > |[NR] PT (0.7) |   7.0|   2.00|     4.39|      1.92|
> > > > > > > |[R] PT (0.1)  |   3.0|   0.10|     0.26|      0.09|
> > > > > > > |[R] PT (0.1)  |   5.0|   0.10|     0.29|      0.10|
> > > > > > > |[R] PT (0.1)  |   7.0|   0.10|     0.28|      0.09|
> > > > > > > |[R] PT (0.1)  |   3.0|   1.00|     1.46|      0.49|
> > > > > > > |[R] PT (0.1)  |   5.0|   1.00|     1.54|      0.45|
> > > > > > > |[R] PT (0.1)  |   7.0|   1.00|     1.73|      0.46|
> > > > > > > |[R] PT (0.1)  |   3.0|   2.00|     2.62|      0.81|
> > > > > > > |[R] PT (0.1)  |   5.0|   2.00|     2.80|      0.80|
> > > > > > > |[R] PT (0.1)  |   7.0|   2.00|     2.88|      1.08|
> > > > > > > |[R] PT (0.3)  |   3.0|   0.10|     0.31|      0.11|
> > > > > > > |[R] PT (0.3)  |   5.0|   0.10|     0.42|      0.12|
> > > > > > > |[R] PT (0.3)  |   7.0|   0.10|     0.47|      0.15|
> > > > > > > |[R] PT (0.3)  |   3.0|   1.00|     1.96|      0.58|
> > > > > > > |[R] PT (0.3)  |   5.0|   1.00|     1.99|      0.56|
> > > > > > > |[R] PT (0.3)  |   7.0|   1.00|     3.31|      0.82|
> > > > > > > |[R] PT (0.3)  |   3.0|   2.00|     4.13|      1.14|
> > > > > > > |[R] PT (0.3)  |   5.0|   2.00|     3.88|      1.15|
> > > > > > > |[R] PT (0.3)  |   7.0|   2.00|     3.67|      1.05|
> > > > > > > |[R] PT (0.5)  |   3.0|   0.10|     0.22|      0.12|
> > > > > > > |[R] PT (0.5)  |   5.0|   0.10|     0.37|      0.13|
> > > > > > > |[R] PT (0.5)  |   7.0|   0.10|     0.50|      0.14|
> > > > > > > |[R] PT (0.5)  |   3.0|   1.00|     1.11|      0.87|
> > > > > > > |[R] PT (0.5)  |   5.0|   1.00|     2.68|      0.79|
> > > > > > > |[R] PT (0.5)  |   7.0|   1.00|     3.89|      0.98|
> > > > > > > |[R] PT (0.5)  |   3.0|   2.00|     1.61|      1.49|
> > > > > > > |[R] PT (0.5)  |   5.0|   2.00|     5.20|      1.51|
> > > > > > > |[R] PT (0.5)  |   7.0|   2.00|     5.32|      1.56|
> > > > > > > |[R] PT (0.7)  |   3.0|   0.10|     0.25|      0.14|
> > > > > > > |[R] PT (0.7)  |   5.0|   0.10|     0.27|      0.18|
> > > > > > > |[R] PT (0.7)  |   7.0|   0.10|     0.33|      0.20|
> > > > > > > |[R] PT (0.7)  |   3.0|   1.00|     0.74|      0.82|
> > > > > > > |[R] PT (0.7)  |   5.0|   1.00|     0.93|      0.67|
> > > > > > > |[R] PT (0.7)  |   7.0|   1.00|     2.41|      1.21|
> > > > > > > |[R] PT (0.7)  |   3.0|   2.00|     1.12|      1.33|
> > > > > > > |[R] PT (0.7)  |   5.0|   2.00|     1.56|      1.47|
> > > > > > > |[R] PT (0.7)  |   7.0|   2.00|     4.40|  **1.98**|

---

> > > > > > > > ### Comment · Reviewer_rki5 · 2022-08-08
> > > > > > > > **solid simulations**
> > > > > > > >
> > > > > > > > Thank you for these additional and interesting simulations. It is very reassuring to see that the proposed method is outperforming R-PT, and is on par with NR-PT, when using the same ZigZag kernel. Indeed, there is the question of whether it is actually competitive with a tuned HMC kernel (ie. the baseline sampler when gradients are available), but I think the proposed simulations are fair and convincing.
> > > > > > > >
> > > > > > > > PS: I have updated my rating.

---

> > > > > > > > > ### Author Response · Authors · 2022-08-09
> > > > > > > > > **Completed simulations**
> > > > > > > > >
> > > > > > > > > Thank you for the updated score. All simulations on the cluster are now complete and have been added to the previous tables. With significantly less tuning our proposed tempering method performs comparably to PT (please pursue the updated tables for reference).

---

### Official Review · Reviewer_77P1 · 2022-07-12

**Rating:** 6
**Confidence:** 3
**Soundness:** 3 good
**Presentation:** 3 good
**Contribution:** 3 good

**Summary:**

The authors show how to use (continuous) simulated tempering in conjunction with a PDMP sampler (in particular, the zig-zag sampler) to sample from multi-modal distributions. In the process that they propose, once the process hits $\beta=1$, it spends an amount of time there that is an exponential rv; this ensures that the marginal distribution of the inverse temperature is a mixture of a continuous distribution on [0,1) and a point mass at 1, which allows obtaining samples directly rather than from importance/rejection sampling. Experiments show significantly improved performance compared with the non-tempered algorithm, as well as how the root-mean-square error and thinning efficiency depends on the value chosen for alpha (parameter controlling how much time the process spends at $\beta=1$).

**Questions:**

* Line 61-62: "this means we can obtain samples from $\pi(x)$ rather than having to resort to importance sampling to correct samples drawn at different temperatures" - in what sense is the proposed approach better than using correction using importance sampling?
* Line 66-67: Can you explain or give a reference for the issue with the HMC sampler?
* Line 156-157: Can't you use importance sampling no matter what \beta is, by using the ratio of the unnormalized distribution at the temperature of the sample and at temperature 1?

Minor notes:

* line 171: "important" sampling -> "importance"


**Limitations:**

Yes.

**Strengths And Weaknesses:**

The authors propose a simple, easy-to-use extension of PDMP (specifically the zig-zag sampler) that significantly improves its performance on multi-modal objectives in simulations. The simulations show how the parameters can be tuned to minimize the error. The paper is generally clear.

However, only the zig-zag sampler was considered in this work. Furthermore, one limitation of the simulated tempering-based approach is the necessity of estimating the normalizing constant (discussed in section 3.4).

---

> ### Author Response · Authors · 2022-08-02
> **Response to reviewer 77P1**
>
> Thank you for your thoughtful review as well as your time and effort.
>
> Regarding the limitation of our work to the Zig-Zag, the paper now includes an additional remark after Theorem 1, detailing how a *tempered* continuous-time sampler may be implemented by augmenting an existing continuous-time sampler with inverse temperature variable $\beta$ that has associated Zig-Zag dynamics. This allows for tempering with alternative dynamics such as Bouncy Particle or Boomerang dynamics with minimal modification.
>
> We briefly address the comments and questions below:
>
> 1. There are several reasons why a pure MCMC as opposed to a combined MCMC and importance sampling method may be preferred, particularly in the case of piecewise deterministic Markov process samplers. Aside from simplicity of a pure-MCMC approach,
>
> - Importance sampling requires additional post-hoc computation. The reason for this is that one needs to choose a uniform discretisation of the simulated Zig-Zag trajectory, and then evaluate the target density at each point to construct the importance weights.
>
> - Importance sampling in general is susceptible to poor performance in high dimensions (see e.g., the discussion in [1], and the references therein).
>
> - Straightforward use of importance sampling approaches in the continuously-tempered requires a marginalized approach (as in [2]), which restricts the choice of pseudo-prior. While importance sampling outside of this setting may be possible, it introduces several other complications (see response to question 3, below).
>
> - The results of our numerical experiments seem to indicate that a pure MCMC approach is preferable to one that is a hybrid with importance sampling.
>
> 2. The issue is simply that standard HMC (and more generally advanced variants such as the No-U-Turn Sampler) are require purely continuous target distributions. Thus, they are not directly capable of sampling a target with a point mass. The new version of the paper has the sentence slightly modified to make the point clearer.
>
> 3. Regarding the use of importance sampling in a general setting. The importance sampling approach discussed in the paper is specifically a marginalised one --- the chain's (stationary) marginal distribution on $x$-space can be computed. The advantage of such an approach is that all samples come from the same proposal distribution and so assigning importance sampling weights is straightforward.
> In contrast, one can not simply reweight each individual sample with respect to the corresponding tempered target, as for each different $\beta$ value, a different proposal distribution is being used (in importance sampling, one needs to divide by the sum of the weights from the proposal distribution, of which there will be many). One way around this is that, in principle, one could use importance sampling by specifying a collection of $\beta$ values and using a separate IS estimator for each (using as samples the x-position of the Zig-Zag process at the precise moment when it crosses an inverse temperature value that is in the specified set).
> While not impossible, how to choose the values and execute such a procedure in a reliable manner is not clear. If multiple temperatures are chosen, then several importance sampling estimators (each with a smaller number of samples than the marginalized approach, and hence potentially more bias) would need to be combined. In short, on top of inheriting some of the other drawbacks of importance sampling discussed, a non-marginalized approach could prove difficult to tune and be a somewhat less elegant solution to a marginalised, or pure-MCMC approach.
>
> [1] Vehtari, A., Simpson, D., Gelman, A., Yao, Y., & Gabry, J. (2015). Pareto smoothed importance sampling. arXiv preprint arXiv:1507.02646.
>
> [2] M. M. Graham and A. J. Storkey. (2017) Continuously tempered Hamiltonian Monte Carlo. In Proceedings
> of the Thirty-Third Conference on Uncertainty in Artificial Intelligence (UAI)

---

> > ### Comment · Reviewer_77P1 · 2022-08-09
> > **Thanks for the reply.**
> >
> > In light of the clarifications and the results from the additional experiments, I'm happy to raise my score to 6.

---

> ### Comment · Area_Chair_G1nh · 2022-08-08
> **Please respond to authors**
>
> Dear 77P1,
>
> as the discussion period is drawing to a close, please at least acknowledge the reply of the authors. If there are any final clarifications to direct at the authors, please do so now.
>
> Best,
> AC

---

### Meta-Review · Area_Chair_G1nh · 2022-08-27

**Recommendation:** Accept
**Confidence:** Certain

**Metareview:**

The authors show how to combine a clever temperature schedule for simulated tempering with a Zig-Zag sampler (though the authors suggest this could be expanded to arbitrary PDMP samplers, it was mentioned that this may not be completely obvious), so that the marginal distribution of the inverse temperature is a mixture of a continuous distr. on [0,1) and a point mass at 1--- allowing taking samples directly and obviating the need for importance/rejection sampling. The theory proves stationarity of the resultant chain, and the original simulations demonstrated improvement over the vanilla non-tempered chain. At the request of the reviewers, the authors also compared to other tempering chains and showed encouraging results (even though as was noted, comparison is a bit hard for some methods, since e.g., some are more suited for parallel computation).


**Award:**

No

---

### Decision · Program_Chairs · 2022-09-14

Accept